# SuperLoss: A Generic Loss
# for Robust Curriculum Learning

**Thibault Castells**
Naver Labs Europe
`thibault.castells@wanadoo.fr`

**Philippe Weinzaepfel**
Naver Labs Europe
`philippe.weinzaepfel@naverlabs.com`

**Jerome Revaud**
Naver Labs Europe
`jerome.revaud@naverlabs.com`

## Abstract

Curriculum learning is a technique to improve a model performance and generalization based on the idea that easy samples should be presented before difficult ones during training. While it is generally complex to estimate *a priori* the difficulty of a given sample, recent works have shown that curriculum learning can be formulated dynamically in a self-supervised manner. The key idea is to somehow estimate the importance (or weight) of each sample directly during training based on the observation that easy and hard samples behave differently and can therefore be separated. However, these approaches are usually limited to a specific task (*e.g.,* classification) and require extra data annotations, layers or parameters as well as a dedicated training procedure. We propose instead a simple and generic method that can be applied to a variety of losses and tasks without any change in the learning procedure. It consists in appending a novel loss function *on top* of any existing task loss, hence its name: the SuperLoss. Its main effect is to automatically downweight the contribution of samples with a large loss, *i.e.* hard samples, effectively mimicking the core principle of curriculum learning. As a side effect, we show that our loss prevents the memorization of noisy samples, making it possible to train from noisy data even with non-robust loss functions. Experimental results on image classification, regression, object detection and image retrieval demonstrate consistent gain, particularly in the presence of noise.

## 1 Introduction

Curriculum learning (CL) [4], a paradigm inspired by the learning process of humans and animals, has recently received a sustained attention [13, 18, 48, 55]. CL is based on the intuitive observation that our learning process naturally starts from easy notions before gradually transitioning to more complex ones. When applied to machine learning, it essentially boils down to designing a sampling strategy, *i.e.* a curriculum, that would present easy samples to the model before harder ones [13, 22]. While this was shown to be effective at improving the model performance and its generalization power in earlier works [4, 22], they were limited to toy datasets and engineered sampling heuristics, leaving unaddressed the general problem of establishing a curriculum on real-world tasks and datasets.

For this reason, a growing line of research has focused on methods able to automatically determine the curriculum without requiring prior knowledge about the task at hand [18, 25, 34, 44, 48, 49]. This is indeed possible as easy and hard samples behave differently during training in terms of their respective loss, allowing them to be somehow discriminated [19, 34, 44]. In this context, CL is accomplished by predicting the easiness of each sample at each training iteration in the form

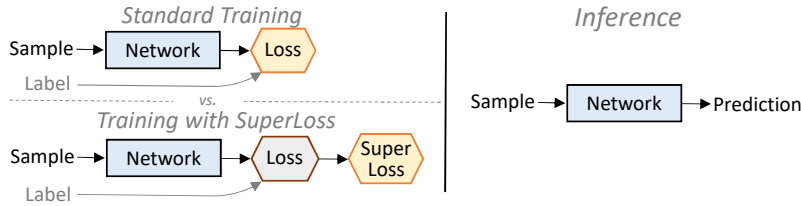

Figure 1: *Top-left:* The classical supervised training. *Bottom-left:* Our approach consists in appending our SuperLoss on top of any existing loss, without changing anything else in the training procedure. Back-propagation now starts from the SuperLoss. *Right:* At test time, no change is required.

of a weight, such that easy samples receive larger weights during the early stages of training and conversely. Another benefit of this type of approach, aside from improving the model generalization, is their resistance to noise. This is due to the fact that noisy samples (*i.e.* with wrong labels) tend to be harder for the model and thus receive smaller weights throughout training [18, 44, 48], effectively discarding them. This side effect makes these methods especially attractive when clean annotated data are expensive while noisy data are widely available and cheap [2, 20, 31, 36, 52]. Existing approaches for automatic CL nevertheless suffer from two important drawbacks that drastically limit their applicability. First, current methods like [6, 18, 19, 28, 48, 55] overwhelmingly focus and specialize on the classification task, even though the principles mentioned above are general and can potentially apply to other tasks. Second, they all require important changes in the training procedure, often requiring dedicated training schemes [1, 6, 8, 16, 28, 55], involving multi-stage training with or without special warm-up periods [13, 18, 19, 23, 39, 60], extra learnable parameters and layers [6, 18, 29, 48, 60, 67] or a clean subset of data [18, 27, 44].

In this paper, we propose instead a simple yet generic approach to dynamic curriculum learning. It is inspired by recent confidence-aware loss functions that yield the capability to jointly learn network parameters and sample weights, or confidences, in a unified framework [38, 46, 48]. As a first contribution, we introduce a novel type of confidence-aware loss function that can transform any task loss into a confidence-aware version. Thanks to an automatic and optimal confidence-setting scheme, it can scale-up to any number of samples and requires no modification of the learning procedure except the insertion of a new loss termed *SuperLoss*, making it broadly applicable. As shown in Figure 1, the SuperLoss is simply plugged *on top* of the original task loss during training, hence its name. Its role is to monitor the loss of each sample during training and to determine the sample contribution dynamically by applying the core principle of curriculum learning. To the best of our knowledge, this is the first time that a task-agnostic approach for curriculum learning without any change in the training procedure is proposed. As a second contribution, we present how the SuperLoss can be applied to various tasks: image classification, deep regression, object detection and image retrieval. As a third contribution, we present empirical evidence that our approach leads to consistent gains when applied on clean and noisy datasets. In particular, we observe large gains in the case of training from noisy data, a typical case for large-scale datasets automatically collected from the web.

## 2 SuperLoss

We first present a family of specialized loss functions that we denote as confidence-aware and which are closely related to CL (Section 2.1). We then derive a generic task-agnostic formulation in Section 2.2 and show how this formulation can be further simplified in the context of CL, yielding the SuperLoss (Section 2.3). Finally, we illustrate in Section 2.4 several applications of our approach.

### 2.1 Confidence-aware loss

A novel type of loss functions, which we denote as confidence-aware, has been recently and independently proposed by several authors for a variety of tasks and backgrounds [21, 38, 46–48]. Consider a dataset $\{(\boldsymbol{x}_i, y_i)\}_{i=1}^N$, where sample $\boldsymbol{x}_i$ has label $y_i$, and let $f(\cdot)$ be a trainable predictor to be optimized using empirical risk minimization. In contrast to traditional loss functions of the form $\ell(f(\boldsymbol{x}_i), y_i)$, a confidence-aware loss function $\ell(f(\boldsymbol{x}_i), y_i, \sigma_i)$ takes an additional learnable parameter $\sigma_i \geq 0$ as input. Such a parameter is associated with each sample $\boldsymbol{x}_i$ and represents the *confidence* or *reliability* of the corresponding prediction $f(\boldsymbol{x}_i)$.

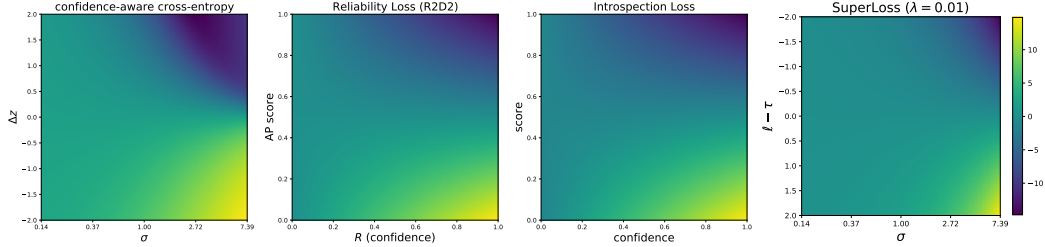

Figure 2: Overall shape of various confidence-aware losses that learn the confidence, with from left to right: confidence-aware cross-entropy [48], reliability loss [46], introspection loss [38], and our proposed SuperLoss. Each plot shows the resulting loss according to the 'correctness' of a particular prediction (y-axis) and the corresponding confidence (x-axis). Blue is smaller and yellow is larger.

The goal of a confidence-aware loss is to handle difficult samples without resorting to heuristics such as using robust versions of the loss, by instead modulating the loss amplitude *w.r.t.* the confidence parameter [38]. We plot several existing confidence-aware loss functions in Figure 2, where the sample confidence $\sigma_i$ is represented on the $x$-axis and a quantity measuring the correctness of the network prediction is represented on the $y$-axis. Interestingly, these losses have practically identical shapes despite the fact that they have been proposed independently for different tasks (*e.g.,* patch matching [46], keypoint detection [38], dynamic CL [48]) and have seemingly unrelated formula (see Supplementary). Regardless of the manner the confidence intervenes in the loss formula, a key property that they noticeably share is that the gradient of the loss *w.r.t.* the network parameters monotonously increases with the confidence, all other parameters staying fixed. Simply put, the left-hand side of the plots (low-confidence area of the loss) is flatter than the right-hand side (high-confidence area). As a consequence, gradient updates towards the model parameters are smaller for samples with lower confidence, which practically amounts to down-weight low-confidence samples during training.

This property makes confidence-aware losses particularly well suited to dynamic CL, as it allows to learn the confidence, *i.e.* weight, of each sample automatically through back-propagation and without further modification of the learning procedure. This principle was recently implemented by Saxena *et al.* [48] for the classification task with a modified confidence-aware cross-entropy loss. As a result, jointly minimizing the network parameters and the confidence parameters, named *data parameters* in [48], via standard stochastic gradient descent leads to accurately estimate the reliability of each prediction, *i.e.* the difficulty of each sample, via the confidence parameter.

### 2.2 A task-agnostic confidence-aware loss function

While the closely-related principles behind dynamic CL and confidence-aware losses appears to be completely generic, we surprisingly find that none of the existing confidence-aware formulations easily generalize to other tasks. For instance, the introspection loss [38] is designed for the retrieval task; the modified cross-entropy from [48] specializes in the classification task; the multi-task loss [21] only handles regression and cross-entropy; etc. We refer to the Supplementary material for more details.

In this work, we propose instead a novel and task-agnostic type of confidence-aware loss function. In contrast to existing confidence-aware losses, it only takes two inputs, namely, the task loss $\ell(f(\boldsymbol{x}_i), y_i)$ (simplified as $\ell_i$ and referred to as input loss in the following) and a confidence parameter $\sigma_i$. We denote this function as $L_\lambda(\ell_i, \sigma_i)$, where $\lambda > 0$ is a regularization hyper-parameter. Even though some confidence-aware losses are built from probabilistic considerations [21, 38, 47], we find it hard to derive a probabilistic framework that would suit all possible types of loss functions. Instead, we consider generic principles and identify three properties that $\mathrm{L}_\lambda(\ell_i, \sigma_i)$ needs to satisfy:

1. **Translation-invariance**. Adding a constant to the input loss should have no effect on $L_\lambda$'s gradient, *i.e.* $\forall K, \ \exists K' \mid \mathrm{L}_\lambda(\ell_i + K, \sigma_i) = K' + \mathrm{L}_\lambda(\ell_i, \sigma_i)$, where $K$ and $K'$ are constant.

2. **Homogeneity**. $L_\lambda$ should have a multiplicative scaling behavior: $\exists \lambda, \lambda' \mid \forall K > 0, \ \mathrm{L}_\lambda(K\ell_i, \sigma_i) = K \, \mathrm{L}_{\lambda'}(\ell_i, \sigma_i)$, where $K$ is a constant. This way we can handle in-

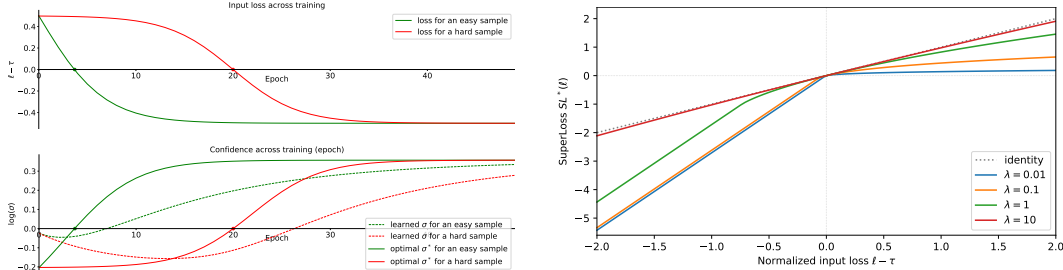

Figure 3: **Left:** We plot on top the typical losses incurred by an easy (green) and a hard (red) sample during training. At bottom, we show (a) their respective confidence when learned via back-propagation using $L_\lambda(\ell_i, \sigma_i)$ (dotted lines) and (b) their optimal confidence $\sigma_\lambda^*(\ell_i)$ (plain lines). In contrast to using the optimal confidence, learning it induces a significant delay between the moment a sample becomes easy (its loss passes under $\tau$) and the moment its confidence becomes greater than 1. **Right:** we show $SL_\lambda(\ell_i)$ as a function of $\ell_i - \tau$ for different $\lambda$. The SuperLoss emphasizes the input loss when $\ell_i < \tau$, and reduces it in the opposite case, thus limiting the impact of hard samples.

put losses of any amplitude, *i.e.* we just need to accordingly rescale the learning rate and $\lambda$.

3. **Generalization**, in the sense that $L_\lambda(\ell_i, \sigma_i)$ should amount to the input loss for a particular confidence $\sigma$, *i.e.* $\exists \sigma \mid L_\lambda(\ell_i, \sigma) = \ell_i + K$, where $K$ is a constant.

Optionally, a convenient aspect is that it should be easily interpretable.

We now propose one of the simplest possible formulation that meets all these criteria, including the interpretability. Similarly to the confidence-aware cross-entropy proposed in [48], it is composed of a loss-amplifying term and a regularization term controlled by the hyper-parameter $\lambda > 0$:

$$L_\lambda(\ell_i, \sigma_i) = (\ell_i - \tau)\,\sigma_i + \lambda\,(\log \sigma_i)^2\,, \tag{1}$$

where $\tau$ is a threshold that ideally separates easy samples from hard samples based on their respective loss. In practice, $\tau$ is empirically estimated as a running average of the input loss during training, thereby trivially satisfying translation-invariance (property 1). Note that a similar thresholding involving extra learnable layers and parameters was proposed to separate easy and hard samples in MentorNet [18]. In certain cases, $\tau$ can also be set as a constant based on prior knowledge on the task loss, but our results suggest that this makes almost no difference compared to using a running average (see Section 3). As for the other properties, homogeneity (property 2) is verified with $\lambda = K\lambda'$ and generalization (property 3) is achieved for $\sigma_i = 1$ as $L_\lambda(\ell_i, 1) = \ell_i - \tau$. We plot the shape of $L_\lambda(\ell_i, \sigma_i)$ in Figure 2 (right): its shape is similar to other specialized confidence-aware losses.

## 2.3 Optimal confidence and SuperLoss

While Saxena *et al.* [48] have shown the benefits of learning the confidence of each sample dynamically via back-propagation, this has several shortcomings. First, it requires one extra learnable parameter $\sigma_i$ per sample, which does not scale for tasks like detection or retrieval where the number of samples can be almost infinite (Section 2.4). Second, learning the confidence naturally induces a delay (*i.e.* the time of convergence), and thus potential inconsistencies between the true status of a sample and its respective confidence, see Figure 3 (left). Third, it adds several hyper-parameters on top of the baseline approach such as the learning rate and weight decay of the secondary optimizer.

Instead of waiting for the confidence parameters to converge, we therefore propose to directly use their converged value at the limit, which only depends on the input loss $\ell_i$:

$$\sigma_\lambda^*(\ell_i) = \arg\min_{\sigma_i} L_\lambda(\ell_i, \sigma_i). \tag{2}$$

As a consequence, the confidence parameters do not need to be learned and are up-to-date with the sample status. The new loss function that we obtain takes a single parameter as input and can therefore simply be appended on top of any given task loss (see Figure 1), hence its name of SuperLoss (SL):

$$SL_\lambda(\ell_i) = L_\lambda\left(\ell_i, \sigma_\lambda^*(\ell_i)\right) = \min_{\sigma_i} L_\lambda\left(\ell_i, \sigma_i\right). \tag{3}$$

As we demonstrate in Section 2.1 of the Supplementary, the optimal confidence $\sigma_\lambda^*(\ell_i)$ from Eq. (2) has a closed-form solution. In practice, we cap the loss to avoid infinite values as follows:

$$\sigma_\lambda^*(\ell_i) = e^{-W\left(\frac{1}{2}\max(-\frac{2}{e},\beta)\right)} \quad \text{with } \beta = \frac{\ell_i - \tau}{\lambda}, \tag{4}$$

where W stands for the Lambert W function. During back-propagation, $\tau$ and $\sigma_\lambda^*(\ell_i)$ are computed from the input loss $\ell_i$ and then treated as constant. We plot our SuperLoss in Figure 3 (right) as a function of the input loss for various $\lambda$. As intended, it amplifies the contribution of easy samples (*i.e.* when $\ell_i < \tau$) while strongly flattening the input loss for hard ones. We show in Section 2.2 of the Supplementary that when the regularization parameter $\lambda$ tends to infinity, the optimal confidence tends to 1, and thus the SuperLoss amounts to the input loss: $\lim_{\lambda \to \infty} \text{SL}_\lambda(\ell_i) = \ell_i - \tau$.

### 2.4 Applications

We now present concrete application cases of the SuperLoss for various tasks.

**Classification.** We straightforwardly plug the Cross-Entropy loss ($\ell^{CE}$) into the SuperLoss: $\text{SL}^{CE}(\boldsymbol{x_i}, y_i) := \text{SL}\left(\ell^{CE}(f(\boldsymbol{x_i}), y_i)\right)$. When specified, we use a fixed threshold for $\tau = \log C$ (where $C$ is the number of classes) as it represents the cross-entropy of a uniform distribution and hence appears to be a natural boundary between correct and incorrect predictions.

**Regression.** Likewise, we can plug a regression loss $\ell^{reg}$ such as the smooth-L1 loss (smooth-$\ell_1$) or the Mean-Square-Error (MSE) loss ($\ell_2$) into the SuperLoss. Note that the range of values for a regression loss drastically differs from the one of the CE loss, but as we pointed out previously, this is not an issue for the SuperLoss thanks to its homogeneity property.

**Object Detection.** We apply the SuperLoss on the box classification component of two object detection frameworks: Faster R-CNN [45] and RetinaNet [33]. Faster R-CNN classification loss is a standard cross-entropy loss $\ell^{CE}$ on which we plug our SuperLoss $\text{SL}^{CE}$. RetinaNet classification loss is a class-balanced focal loss (FL): $\ell^{FL}(\boldsymbol{p}, y) = -\alpha_y(1 - \boldsymbol{p}_y)^\gamma log(\boldsymbol{p}_y)$ with $\boldsymbol{p}$ the probabilities predicted by the network for each box obtained with a softmax on the logits $\boldsymbol{z} = f(\boldsymbol{x})$, $\gamma \geq 0$ a focusing hyper-parameter and $\alpha_y$ a class-balancing hyper-parameter. We point out that, in contrast to classification, object detection typically deals with an enormous number of negative detections, for which it is infeasible to store or learn individual confidences. In contrast to approaches that learn a separate weight per sample like [48], our method estimates the confidence of positive and negative detections on the fly from their individual loss.

**Retrieval/metric learning.** We apply our SuperLoss to image retrieval using the contrastive loss [14], which was shown to be one of the most effective losses for metric learning [37]. In this case, the training set $\{(\boldsymbol{x}_i, \boldsymbol{x}_j, y_{ij})\}_{i,j}$ is composed of pairs of samples labeled either positively ($y_{ij} = 1$) or negatively ($y_{ij} = 0$). The goal is then to learn a latent representation where positive pairs lie close whereas negative pairs are far apart. The contrastive loss is composed of two losses: $\ell_+^{CL}(f(\boldsymbol{x}_i), f(\boldsymbol{x}_j)) = [\|f(\boldsymbol{x}_i) - f(\boldsymbol{x}_j)\|]_+$ for positive pairs and $\ell_-^{CL}(f(\boldsymbol{x}_i), f(\boldsymbol{x}_j)) = [m - \|f(\boldsymbol{x}_i) - f(\boldsymbol{x}_j)\|]_+$ for negative pairs where $m > 0$ is a margin (we assume a null margin for positive pairs as is common [43] and $[\,.\,]_+$ denotes the positive component). We apply the SuperLoss on top of each of the two losses, *i.e.* with two independent thresholds $\tau_+$ and $\tau_-$, but still sharing the same regularization parameter $\lambda$ for simplicity:

$$SL_\lambda^{CL}(f(\boldsymbol{x}_i), f(\boldsymbol{x}_j), y_{ij}) = \begin{cases} SL_\lambda\left(\ell_+^{CL}(f(\boldsymbol{x}_i), f(\boldsymbol{x}_j))\right) & \text{if } y_{ij} = 1, \\ SL_\lambda\left(\ell_-^{CL}(f(\boldsymbol{x}_i), f(\boldsymbol{x}_j))\right) & \text{if } y_{ij} = 0. \end{cases} \tag{5}$$

The same strategy can be applied to other metric learning losses such as the triplet loss [56]. As for object detection, we note that other approaches that explicitly learn or estimate the importance of each sample cannot be applied to metric learning because (a) the number of potential pairs or triplets is enormous, making intractable to store their confidence in memory; and (b) only a small fraction of them is seen at each epoch, which prevents the accumulation of enough evidence.

## 3 Experimental results

After describing our experimental protocol in Section 3.1, we evaluate the SuperLoss for regression (Section 3.2), image classification (Section 3.3), object detection (Section 3.4) and image retrieval (Section 3.5) each time in clean and noisy conditions.

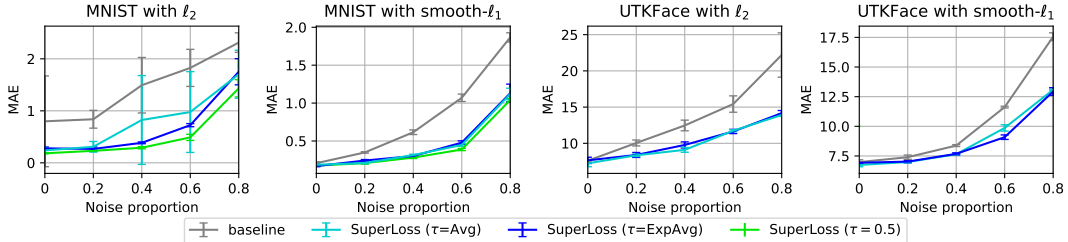

Figure 4: Performance (MAE) on digit regression and human age regression as a function of noise proportion, for a robust loss (smooth-$\ell_1$) and a non-robust loss ($\ell_2$).

## 3.1 Experimental protocol

We refer to the model trained with the original task loss as the baseline. Our protocol is to first train the baseline and tune its hyper-parameters (*e.g.,* learning rate, weight decay, etc.) using held-out validation for each noise level. For a fair comparison between the baseline and the SuperLoss, we train the model with the SuperLoss using the same hyper-parameters. Unlike most existing works (*e.g.,* [6]), we do not need special warm-up periods or other tricks. Hyper-parameters specific to the SuperLoss (regularization $\lambda$ and loss threshold $\tau$) are either fixed or tuned using held-out validation or cross-validation. More specifically, we experiment with three options for $\tau$: (1) a global average of the loss so far, denoted as 'Avg'; (2) an exponential running average with a fixed smoothing parameter $\alpha = 0.9$, denoted as 'ExpAvg'; or (3) a fixed value given by prior knowledge on the task at hand. Similar to SELF [6], we smooth the input loss $\ell_i$ individually for each sample using exponential averaging with $\alpha' = 0.9$, as it makes the training more stable. This strategy is only applicable for limited size datasets; for metric learning or object detection, we do not use it.

## 3.2 Regression

We first evaluate our SuperLoss on digit regression on MNIST and human age regression on UTKFace, with both a robust loss (smooth-$\ell_1$) and a non-robust one ($\ell_2$), and with different noise levels.

**Digit regression.** We perform a toy regression experiment on MNIST [26] by considering the original digit classification problem as a regression problem. Specifically, we set the output dimension of LeNet [26] to 1 instead of 10 and train it using a regression loss for 20 epochs using SGD (Stochastic Gradient Descent). We cross-validate the hyper-parameters of the baseline for each loss and noise level. Typically, $\ell_2$ prefers a lower learning rate compared to smooth-$\ell_1$. For the SuperLoss, we experiment with a fixed threshold $\tau = 0.5$ as it is the acceptable bound for regressing the right integer.

**Age regression.** We experiment on the larger UTKFace dataset [70] which consists of 23,705 aligned and cropped face images, randomly split into 90% for training and 10% for testing. Races, genders and ages (between 1 to 116 years old) widely vary and are represented in imbalanced proportions, making the age regression task challenging. We use a ResNet-18 model (with a single output) initialized on ImageNet as predictor and train for 100 epochs using SGD. Likewise, we cross-validate the hyper-parameters for each loss and noise level. Because it is not clear which fixed threshold would be optimal for this task, we do not use a fixed threshold in the SuperLoss.

**Results.** To evaluate the impact of noise when training, we generate it artificially using a uniform distribution between 1 and 10 for digits and between 1 and 116 for ages. We report the mean absolute error (MAE) aggregated over 5 runs for both datasets and both losses with varying noise proportions in Figure 4. Models trained using the SuperLoss consistently outperform the baseline by a significant margin, regardless of the noise level or the $\tau$ threshold. This is particularly true when the network is trained with a non-robust loss ($\ell_2$), suggesting that the SuperLoss makes a non-robust loss more robust. Even when the baseline is trained using a robust loss (smooth-$\ell_1$), the SuperLoss still significantly reduces the error (*e.g.,* from $17.56 \pm 0.33$ to $13.09 \pm 0.05$ on UTKFace at 80% noise). Note that the two losses have drastically different ranges of amplitudes depending on the task (*e.g.,* $\ell_2$ for age regression typically ranges in $[0, 10000]$ while smooth-$\ell_1$ for digit regression ranges in $[0, 10]$).

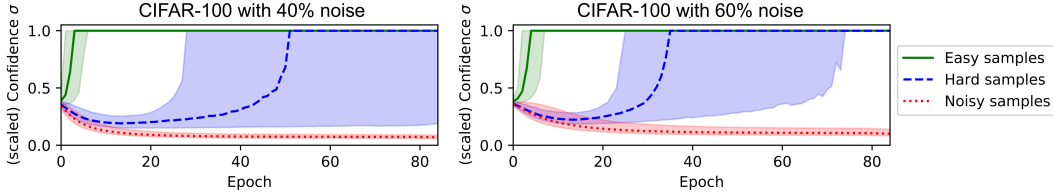

Figure 5: Evolution of the normalized confidence $\sigma^*$ from Equation (2) during training (median value and 25-75% percentiles). We arbitrarily define hard samples as correct samples failing to reach high confidence within the first 20 epochs of training.

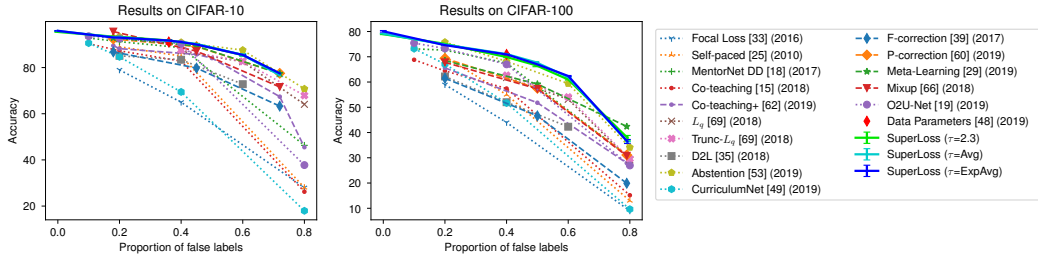

Figure 6: Accuracy on CIFAR-10 and CIFAR-100 as a function of the proportion of noise for our SuperLoss and the state of the art.

### 3.3 Image classification

We next evaluate our SuperLoss for the image classification task on CIFAR-10, CIFAR-100 and WebVision.

**CIFAR-10** and **CIFAR-100** [24] consist of 50K training and 10K test images belonging to $C = 10$ and $C = 100$ classes respectively. We train a WideResNet-28-10 model [63] with the SuperLoss, strictly following the experimental settings and protocol from Saxena *et al.* [48] for comparison purpose. We set the regularization parameter to $\lambda = 1$ for CIFAR-10 and to $\lambda = 0.25$ for CIFAR-100. We plot in Figure 5 the evolution of the confidence $\sigma_\lambda^*$ from Equation (2) for easy, hard and noisy samples. As training progresses, noisy and hard samples get more clearly separated.

We report our results (averaged over 5 runs) for different proportions of corrupted labels in Figure 6, as well as results from the state of the art. Once again, we observe very similar performance regardless of $\tau$ (either fixed to $log(C)$ or using automatic averaging). On clean datasets, the SuperLoss slightly improves over the baseline (*e.g.,* from $95.8\% \pm 0.1$ to $96.0\% \pm 0.1$ on CIFAR-10) even though the performance is quite saturated. In the presence of symmetric noise, the SuperLoss performs better or on par compared to all recent approaches that we are aware of, at the exception of [6, 28]. In particular, our method slightly outperforms the confidence-aware loss proposed by Saxena *et al.* [48] in fair settings, confirming that confidence parameters indeed do not need to be learned. For instance, using global averaging for $\tau$, our method obtains $91.55\% \pm 0.33$ and $71.05\% \pm 0.08$ accuracy under 40% label noise on CIFAR-10 and CIFAR-100 respectively, compared to $91.10\% \pm 0.70$ and $70.93\% \pm 0.15$ for [48]. Finally, note that our method outperforms much more complex and specialized approaches, even though we do not particularly target classification, nor require any change in the network, nor use a special training procedure.

**WebVision** [31] is a large-scale dataset of 2.4 million images with $C = 1000$ classes, automatically gathered from the web by querying search engines with the class names. It thus inherently contains a significant level of noise. We follow [48] and train a ResNet-18 model using SGD for 120 epochs with a weight decay of $10^{-4}$, an initial learning rate of 0.1, divided by 10 at 30, 60 and 90 epochs. The regularization parameter for the SuperLoss is set to $\lambda = 0.25$ and we use a fixed threshold for $\tau = \log(C)$. The final accuracy is $66.7\% \pm 0.1$ which represents a consistent gain of +1.2% (aggregated over 4 runs) compared to the baseline ($65.5\% \pm 0.1$). We point out that this gain is for free as the SuperLoss does not require any change in terms of training time or engineering efforts.

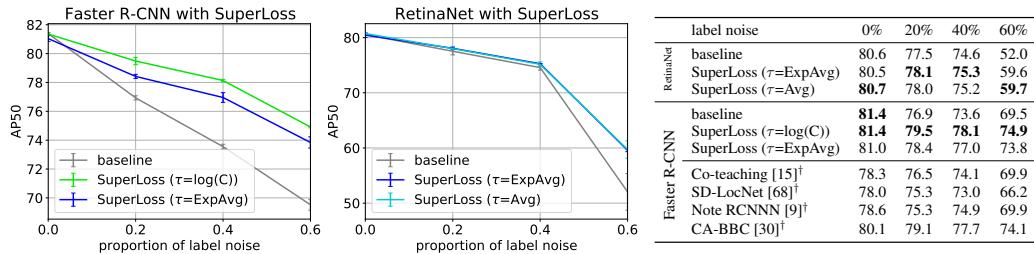

Figure 7: AP50 on Pascal VOC when using the SuperLoss for object detection with Faster R-CNN (*left*) and RetinaNet (*middle*), we report mean and standard deviation over 3 runs. The right table shows a comparison to the state of the art (AP50 metric). [†] denotes numbers reported from [30].

| | label noise | 0% | 20% | 40% | 60% |
|---|---|---|---|---|---|
| RetinaNet | baseline | 80.6 | 77.5 | 74.6 | 52.0 |
| | SuperLoss ($\tau$=ExpAvg) | 80.5 | **78.1** | **75.3** | 59.6 |
| | SuperLoss ($\tau$=Avg) | **80.7** | 78.0 | 75.2 | **59.7** |
| Faster R-CNN | baseline | **81.4** | 76.9 | 73.6 | 69.5 |
| | SuperLoss ($\tau$=log(C)) | **81.4** | **79.5** | **78.1** | **74.9** |
| | SuperLoss ($\tau$=ExpAvg) | 81.0 | 78.4 | 77.0 | 73.8 |
| | Co-teaching [15][†] | 78.3 | 76.5 | 74.1 | 69.9 |
| | SD-LocNet [68][†] | 78.0 | 75.3 | 73.0 | 66.2 |
| | Note RCNNN [9][†] | 78.6 | 75.3 | 74.9 | 69.9 |
| | CA-BBC [30][†] | 80.1 | 79.1 | 77.7 | 74.1 |

## 3.4 Object detection

We perform experiments for the object detection task on Pascal VOC [7] and its noisy version from [30] where symmetric label noise is applied to 20%, 40% or 60% of the instances. We use two object detection frameworks from detectron2[1]: Faster R-CNN [45] and RetinaNet [33]. Figure 7 shows the standard AP50 metric for varying levels of noise using the standard box classification loss or the SuperLoss (more metrics are in the Supplementary). While the baseline and the SuperLoss are on par on clean data, the SuperLoss again significantly outperforms the baseline in the presence of noise. For instance, the performance drop at 60% of label noise is reduced by 5 points for Faster R-CNN, and by 9 points for Retina-Net. For $\tau$, we observe a slight edge for $\tau = log(C)$ with Faster R-CNN. The same fixed threshold makes no sense for RetinaNet as it does not rely on cross-entropy loss, but we observe that global and exponential averaging perform similarly. In Figure 7 (right), we compare our method to some state-of-the-art noise-robust approaches [9, 15, 30, 68]. Once again our simple and generic SuperLoss outperforms other approaches leveraging complex strategies to identify and/or correct noisy samples.

## 3.5 Image retrieval

We evaluate the SuperLoss on the image retrieval task using the Revisited Oxford and Paris benchmark [42]. To train our method, we use the large-scale Landmarks dataset [2] that is composed of about 200K images (divided into 160K/40K for training/validation) gathered semi-automatically using search engines. The fact that a cleaned version of the same dataset (released in [12]) comprises about 4 times less images gives a rough idea of the tremendous amount of noise it contains, and of the subsequent difficulty to leverage this data using standard loss functions. In order to establish a meaningful comparison we also experiment with the cleaned dataset [12] which comprises 42K training and 6K validation images. Following [12], we refer to these datasets as Landmarks-full and Landmarks-clean. As retrieval model, we use ResNet-50 with Generalized-Mean (GeM) pooling [43] and a contrastive loss [14][2]. When training on Landmarks-clean, the default hyper-parameters from [42] for the optimizer and the hard-negative mining procedure gives excellent results (*i.e.* 100 epochs, learning rate of $10^{-6}$ with exponential decay of $\exp(-1/100)$, 2000 queries per epoch and 20K negative pool size). In contrast, they lead to poor performance when training on Landmarks-full. We thus retune the hyper-parameters for the baseline on the validation set of Landmarks-full and find that reducing the hard negative mining hyper-parameters is important (200 queries and pool size of 500 negatives). In all cases, we train the SuperLoss with the same settings than the baseline using global averaging for $\tau$. At test time, we follow [42] and use multiple scales and descriptor whitening.

**Results**. We report the mean Average Precision (mAP) in Table 1. On clean data, the SuperLoss has minor impact. However, it enables an impressive performance boost on noisy data (Landmarks-full), overall outperforming the baseline trained using clean data. This result shows that our SuperLoss makes it possible to train a model from a large automatically-collected dataset with a better performance than from a manually labelled subset. We also include state-of-the-art results trained and evaluated with identical code [42] at the end of Table 1. We perform slightly better than ResNet-

Table 1: Image retrieval results (mAP) for different training sets and losses. Hard-neg indicates the couple of hyper-parameters (query size, pool size) used for hard-negative mining.

| Network + pooling | Training set | Loss | Hard-neg | ROxf (M) | ROxf (H) | RPar (M) | RPar (H) | Avg |
|---|---|---|---|---|---|---|---|---|
| ResNet-50+GeM | Landmarks-clean | contrastive | 2K,20K | 61.1 | 33.3 | 77.2 | 57.2 | 57.2 |
| | | SuperLoss | 2K,20K | 61.3 | 33.3 | 77.0 | 57.0 | 57.2 |
| | Landmarks-full | contrastive | 2K,20K | 41.9 | 17.5 | 65.0 | 39.4 | 41.0 |
| | | contrastive | 200,500 | 54.4 | 28.7 | 72.6 | 50.2 | 51.4 |
| | | SuperLoss | 200,500 | 62.7 | 38.1 | 77.0 | 56.5 | 58.6 |
| ResNet-101+GeM [42] | SfM-120k (clean) | contrastive | 2K,20K | 65.4 | 40.1 | 76.7 | 55.2 | 59.3 |

101+GeM on RParis despite the deeper backbone and the fact that it is trained on SfM-120k, a clean dataset of comparable size requiring a complex and expensive procedure to collect [43].

# 4   Related work

**Curriculum Learning**, a technique inspired by the learning process of humans and animals [50], was introduced in a machine learning context in [4]. The key idea is to feed training samples to the learner in order of increasing difficulty, just like humans naturally learn easier concepts before more complex ones. This was shown to improve the speed of convergence and the quality of the local minima obtained [3, 4, 13, 22, 49]. In these works, the order is determined prior to the training, leading to potential inconsistencies between the fixed curriculum and the model being learned. To remedy this, Kumar *et al.* [25] proposed the concept of self-paced learning where the curriculum is constructed without supervision in a dynamic way to adjust to the pace of the learner. This seminal concept has inspired many variations in diverse domains like classification [5, 40, 48], matrix factorization [17, 71], clustering [10, 57] and object [51, 64, 65] / face detection [32, 58]. Recent works have focused towards learning-based self-paced approaches where the curriculum is performed by learning to weight samples, either explicitly [18, 34, 44, 49, 67] or implicitly [48]. This latter work [48] hinges upon a modified cross-entropy loss modulated by additional data parameters representing the difficulty of each sample. Our SuperLoss can be seen as a generalized and simplified version of it without the need for extra data parameters.

**Learning on noisy data** is a closely related topic, due to the inherent hardness of noisy samples. In this context, CL turns out to be particularly appropriate as it automatically downweights samples based on their difficulty, effectively discarding noisy samples [18, 34, 44, 61]. The recent Curriculum Loss [34], for instance, adaptively selects samples for model training, avoiding noisy samples that have a larger loss. Although not strictly related to CL, a series of works leveraging similar principle has been proposed. O2U-Net [19] distinguishes correct samples from noisy ones by monitoring their loss while varying the learning rate. In [1, 28], the per-sample loss distribution is modeled with a bi-modal mixture model used to dynamically divide the training data into clean and noisy sets. Ensembling methods [6, 9] are also popular to prevent the memorization of noisy samples. For instance, SELF [6] progressively filters samples from easy to hard ones at each epoch, which can be viewed as CL. Co-teaching [15] and similar methods [28, 54, 62] train two semi-independent networks that exchange information about noisy samples to avoid their memorization. However, these approaches are developed specifically for a given task (*e.g.,* classification) and hardly generalize to other tasks. Furthermore, they require a dedicated training procedure which can be cumbersome. Some other approaches focus on designing robust loss functions [11, 15, 33, 41, 59, 69], but they again specialize on a single task. Our SuperLoss is inspired by the same principle but can be easily plugged on top of any loss without any change in the learning procedure.

# 5   Conclusion

We have introduced the SuperLoss, a simple task-agnostic loss function which can be plugged on top of any loss during training. It is a confidence-aware loss function where the optimal confidence can be expressed in a closed-form solution. Our results on a variety of tasks show that using the SuperLoss implicitly performs curriculum learning, leading to inherent noise robustness properties.

## Broader Impact

Our approach can be used on top of any loss, and thus applied to various tasks: it basically applies the principles of automatic curriculum learning to any learning problem. The main benefit is that it allows to train models that will perform better, especially in the case where training data are corrupted by noise. Note that this point might actually be considered as extremely positive given the enormous annotation efforts necessary to build very large-scale datasets and previously thought as unavoidable to reach high performance. Having to annotate a large-scale dataset might be a real barrier for new players to enter into the business for an existing task or for developing new services, because of both the financial aspects and the time it would take. Besides, the annotation effort is most of the time accomplished in poor working conditions. It is often not even considered as a salaried job, thus preventing from social advantages of real jobs including a minimal decent salary. For these reasons and thanks to the simple and generic nature of our approach, we believe in its wide adoption and usage by both research and industrial communities.

Concerning the downsides, we first generally note that our main benefit, training a better model, is also applicable to tasks that may have a negative impact on the society. We also note that our SuperLoss generally has trouble to significantly outperform a baseline approach when training from a clean dataset of comparable size. For applications that require very high precision (*e.g.,* for medical diagnosis), manual annotations would still be more than recommended, and even needed, compared to adopting our solution for noise-resistant training.

## Acknowledgments and Disclosure of Funding

None of the authors received any third party funding nor any third party support during the last 36 months prior to this submission. There was also no additional revenues related to this work.

## Footnotes

[1]`https://github.com/facebookresearch/detectron2`

[2]`https://github.com/filipradenovic/cnnimageretrieval-pytorch`

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
