[Supplementary Material]

# Supplementary Material for
# SuperLoss: A Generic Loss
# for Robust Curriculum Learning

**Thibault Castells**
Naver Labs Europe
thibault.castells@wanadoo.fr

**Philippe Weinzaepfel**
Naver Labs Europe
philippe.weinzaepfel@naverlabs.com

**Jerome Revaud**
Naver Labs Europe
jerome.revaud@naverlabs.com

In this supplementary material, we first provide some details on the different confidence-aware loss functions mentioned in Section 1 of the main paper. We then demonstrate some key mathematical properties of the SuperLoss in Section 2. Finally, we provide a more detailed version of the experimental results in Section 3.

## 1   Confidence-aware Losses

In Section 2.1 of the main paper, we give several examples of existing confidence-aware loss functions. We now detail their formula and provide some explanation on how they are represented in Figure 2 of the main paper.

**The introspection loss [15]** was introduced in the context of keypoint matching between different class instances. It can be rewritten from the original formulation [15] in a more compact form as:

$$\ell_{introspection}(s, y, \sigma) = \log\left(\frac{\exp(\sigma) - 1}{\sigma}\right) - \sigma y s, \qquad (1)$$

where $s \in [0, 1]$ is the similarity score between two keypoints computed as a dot-product between their representation, $y \in \{-1, 1\}$ is the ground-truth label for the pair and $\sigma > 0$ is an input-dependent prediction of the reliability of the two keypoints. In Figure 2 of the main paper, we plot $\sigma$ on the $x$-axis and the similarity score $s$ on the $y$-axis, assuming a positive label $y = 1$. Note that this loss hardly generalizes to other tasks as it is specially designed to handle similarity scores in the range $[0, 1]$ with binary labels.

**Reliability loss from R2D2 [22].**   Introduced in the context of robust patch detection and description, the reliability loss from R2D2 serves to jointly learn a patch representation along with its reliability (*i.e.* a confidence score for the quality of the representation), which is also an input-dependent output of the network [22]. It is formulated as:

$$\ell_{R2D2}(z, y, \sigma) = \sigma(1 - AP(z, y)) + \frac{1 - \sigma}{2}, \qquad (2)$$

where $z$ represents a patch descriptor, $y$ its label and $\sigma \in [0, 1]$ its reliability [1]. The score for the patch is computed in the loss in term of differentiable Average-Precision (AP). In Figure 2 of the main paper, we plot $\sigma$ on the $x$-axis and $AP(z, y)$ on the $y$-axis. We identify two shortcomings with this formulation of a confidence-aware loss function. First, the reliability $\sigma$ is not an unconstrained

variable (it is bounded between 0 and 1), making it difficult to regress in practice. Second, due to the lack of regularization, the optimal reliability is actually either 0 or 1, depending on whether $AP(z, y) < 0.5$ or not. In other words, for a given fixed $AP(z, y) < 0.5$, the loss is minimized by setting $\sigma = 0$ and vice versa, *i.e.* it only encourages the reliability to take extreme values.

**Modified cross-entropy loss [23].** Saxena *et al.* [23] have introduced for classification a tempered version of the cross-entropy loss where a sample-dependent temperature scales logits before computing the softmax:

$$\ell_{DataParams}(z, y, \sigma) = -\log\left(\frac{\exp(\sigma z_y)}{\sum_j \exp(\sigma z_j)}\right), \tag{3}$$

where $z \in \mathbb{R}^C$ are the logits for a given sample ($C$ is the number of classes), $y \in \{1, \dots, C\}$ its ground-truth class and $\sigma > 0$ its confidence (*i.e.* inverse of the temperature in [23])[2]. Interestingly, this loss transforms into a robust 0-1 loss (*i.e.* step function) when the confidence tends to infinity:

$$\lim_{\sigma \to +\infty} \ell_{DataParams}(z, y, \sigma) = \begin{cases} 0 & \text{if } z_y > \max_j z \\ +\infty & \text{otherwise.} \end{cases} \tag{4}$$

Similarly to the SuperLoss, a regularization term equal to $\lambda \log(\sigma)^2$ is added to the loss to prevent $\sigma$ to blow up. In Figure 2 of the main paper, we assume for the sake of representation a two-class problem (hence $z \in \mathbb{R}^2$) and $y = 1$ and plot $\sigma$ on the $x$-axis and $\Delta z = z_1 - z_0$ on the $y$-axis. Because this loss cannot be negative, we plot the log of the loss. While this loss handles well the case of classification, we find that, similarly to previous confidence-aware losses, it hardly generalizes to other tasks.

**Multi-task loss [7].** Not represented in Figure 2 of the main paper but following a similar idea, a scheme to automatically learn the relative weight of each loss in a multi-task context has been proposed by Kendall *et al.* [7]. The intuition is to model the network prediction as a probabilistic function that depends on the network output and an uncontrolled homoscedastic uncertainty. Then, the log likelihood of the model is maximized as in maximum likelihood inference. It leads to the following minimization objective, defined according to several task losses $\{\ell_1, \dots, \ell_n\}$ with their associated uncertainties $\{\sigma_1, \dots, \sigma_n\}$ (*i.e.* inverse confidences):

$$\ell_{multitask}(\ell_1, \dots, \ell_n, \sigma_1, \dots, \sigma_n) = \sum_{i=1}^n \frac{\ell_i}{2\sigma_i^2} + \log \sigma_i, \tag{5}$$

In practice, the confidence is learned via an exponential mapping $s = \log \sigma^2$ to ensure that $\sigma > 0$. Note that this approach makes the implicit assumption that task losses are positive with a minimum $\min \ell_i = 0 \ \forall i$, which is not guaranteed in general. In the case where one of the task loss would be negative, nothing would indeed prevent the multi-task loss to blow up to $-\infty$.

**SuperLoss.** For the sake of exhaustiveness, we recall the formula of the proposed SuperLoss:

$$\text{SL}_\lambda^*(\ell, \sigma) = \sigma(\ell - \tau) + \lambda (\log \sigma)^2 \tag{6}$$

where the loss $\ell$ and the confidence $\sigma$ corresponds to an individual training sample. Similarly to most of the aforementioned confidence-aware losses, we use an exponential mapping $\sigma = e^c$ to ensure that $\sigma > 0$ (see also Section 2 of this Supplementary). While we note that our loss is similar in principle to the multi-task loss [7], we point out some important differences. First, the SuperLoss is applied individually for each sample at the lowest level; on the contrary it is applied at the highest multi-task level in [7]. Second, the SuperLoss includes an additional regularization term $\lambda$ that allows to handle losses of different amplitudes and different levels of noise in the training set (see Section 2 below). Lastly, the SuperLoss makes no assumption on the range and minimum value of the loss, thereby introducing a dynamic threshold $\tau$ and a squared log of the confidence for the regularization.

Figure 1: We plot all confidence-aware losses described in Section 1 on the first row. On the second row, we show for each loss the absolute value of its gradient with respect to the parameter on the $y$ axis. This represents the impact of the loss on the network parameters at any given position on the plot. At the exception of [23] (locally), the gradient monotonously increases with the confidence in all cases. Blue (yellow) is smaller (larger) and black lines correspond to lines of equal value.

**Confidence-aware loss and sample weighting.** The confidence directly corresponds to the weighting of the sample losses in the SuperLoss, making it easily interpretable. However, the relation between confidence and sample weighting is not necessarily obvious for the other aforementioned losses. Regardless of the manner the confidence intervenes in the formula, the key property for a confidence-aware loss is that the gradient of the loss *w.r.t.* the network parameters should monotonously increase with the confidence, all other parameters staying fixed. This is illustrated in the second row of Figure 1. This implies that gradient updates towards the model parameters are smaller for samples with lower confidence, which is effectively equivalent of down-weighting low-confidence samples during training.

## 2   More details on the SuperLoss

### 2.1   Closed-form formula for the SuperLoss

To compute the closed-form $\mathrm{SL}^*(\ell)$ of the SuperLoss defined in Section 2.2 of the main paper, it is necessary to find the confidence value $\sigma^*(\ell)$ that minimizes $\mathrm{SL}(\ell, \sigma)$ for a given loss $\ell$. We first rewrite the equation using an exponential mapping for the confidence:

$$
\begin{aligned}
\mathrm{SL}^*(\ell) &= \min_{\sigma}\ \sigma(\ell - \tau) + \lambda(\log \sigma)^2 \\
&= \min_{x}\ e^x(\ell - \tau) + \lambda x^2 \\
&= \min_{x}\ \beta e^x + x^2,
\end{aligned}
\tag{7}
$$

where

$$
\lambda > 0, \quad \beta = \frac{\ell - \tau}{\lambda} \text{ and } \sigma = e^x.
$$

The function to minimize admits a global minimum in the case where $\beta \geq 0$, as it is the sum of two convex functions. Otherwise, due to the negative exponential term it diverges towards $-\infty$ when $x \to +\infty$. However, in the case where $\beta_0 < \beta < 0$ with $\beta_0 = -2/e$, the function admits a single local minimum, located in $x \in [0, 1]$ (see below), which corresponds to the value the confidence would converge to assuming that it initially starts at $\sigma = 1$ ($x = 0$) and move continuously by infinitesimal displacements. In the case where it exists (i.e., when $\beta_0 < \beta$), the position of the minimum is given by solving the root of the derivative:

$$\frac{\partial}{\partial x}\left(\beta e^x + x^2\right) = 0$$
$$\iff \beta e^x + 2x = 0$$
$$\iff \beta e^x = -2x$$
$$\iff \frac{\beta}{2} = -x\exp(-x) \tag{8}$$

This is an equation of the form $z = ye^y$ with $z \in \mathbb{R}$ and $y \in \mathbb{R}$, a well-known problem having $y = W(z)$ for solution, where W stands for the Lambert W function. The closed form for $x$ in the case where $\beta_0 < \beta$ is thus generally given by:

$$x = -W(\frac{\beta}{2})$$
$$\iff \log \sigma^* = -W(\frac{\beta}{2})$$
$$\iff \sigma^* = e^{-W(\frac{\beta}{2})} \tag{9}$$

Due to the fact that the Lambert W function is monotonously increasing, the minimum is located in $x \in [-\infty, 0]$ when $\beta \geq 0$ and in $x \in [0, 1]$ when $\beta_0 < \beta < 0$. Although the Lambert W function cannot be expressed in term of elementary functions, it is implemented in most modern math libraries[3]. In our case, we use a precomputed piece-wise approximation of the function, which can be easily implemented on GPU in PyTorch using the `grid_sample()` function. In the case where $\beta \leq \beta_0$, we cap the optimal confidence at $\sigma^* = e^{-W(\beta_0/2)} = e$. In summary,

$$\sigma^*_\lambda(\ell) = e^{-W\left(\frac{1}{2}\max(\beta_0,\beta)\right)} \quad \text{with } \beta = \frac{\ell - \tau}{\lambda}. \tag{10}$$

## 2.2 Effect of large regularization

**Lemma 1.** *The SuperLoss becomes equivalent to the original input loss when $\lambda$ tends to infinity.*

*Proof.* As a corollary of Equation (10), we obtain:

$$\lim_{\lambda \to +\infty} \sigma^*_\lambda(\ell) = e^{-W(0)} = 1, \tag{11}$$

hence

$$\lim_{\lambda \to +\infty} \mathrm{SL}^*_\lambda(\ell) = \lim_{\lambda \to +\infty} \sigma^*_\lambda(\ell)(\ell - \tau) + \lambda\left(\log \sigma^*_\lambda(\ell)\right)^2, \tag{12}$$
$$= \lim_{\lambda \to +\infty} \ell - \tau + \lambda W\left(\frac{\ell - \tau}{2\lambda}\right)^2$$
$$= \lim_{\lambda \to +\infty} \ell - \tau + \frac{(\ell - \tau)^2}{4\lambda}$$
$$= \ell - \tau.$$

Since $\tau$ is considered constant, the SuperLoss is equivalent to the input loss $\ell$ at the limit. $\qquad\square$

## 2.3 Homogeneity of the SuperLoss

One important property of the SuperLoss is its homogeneity, meaning that it can handle an input loss of any given range and thus any kind of tasks. More specifically this means that the shape of the SuperLoss stays exactly the same, up to a constant scaling factor $\gamma > 0$, when the input loss and the regularization parameter are both scaled by the same factor $\gamma$. In other words, it suffices to scale accordingly the regularization parameter and the learning rate to accommodate for an input loss of any given amplitude.

Table 1: Regression results in term of mean absolute error (MAE) aggregated over 5 runs (mean ± standard deviation) on the task of digit regression on the MNIST dataset [9].

| Input Loss | Method | Proportion of noise | | | | |
| --- | --- | --- | --- | --- | --- | --- |
| | | 0% | 20% | 40% | 60% | 80% |
| MSE ($\ell_2$) | Baseline | $0.80 \pm 0.87$ | $0.84 \pm 0.17$ | $1.49 \pm 0.53$ | $1.83 \pm 0.36$ | $2.31 \pm 0.19$ |
| | SuperLoss | $\mathbf{0.18 \pm 0.01}$ | $\mathbf{0.23 \pm 0.01}$ | $\mathbf{0.29 \pm 0.02}$ | $\mathbf{0.49 \pm 0.06}$ | $\mathbf{1.43 \pm 0.17}$ |
| smooth-$\ell_1$ | Baseline | $0.21 \pm 0.02$ | $0.35 \pm 0.01$ | $0.62 \pm 0.03$ | $1.07 \pm 0.05$ | $1.87 \pm 0.06$ |
| | SuperLoss | $\mathbf{0.18 \pm 0.01}$ | $\mathbf{0.21 \pm 0.01}$ | $\mathbf{0.28 \pm 0.01}$ | $\mathbf{0.39 \pm 0.01}$ | $\mathbf{1.04 \pm 0.02}$ |

Table 2: Regression results in term of mean absolute error (MAE) aggregated over 5 runs (mean ± standard deviation) on the task of human age regression on the UTKFace dataset [30].

| Input Loss | Method | Proportion of noise | | | | |
| --- | --- | --- | --- | --- | --- | --- |
| | | 0% | 20% | 40% | 60% | 80% |
| MSE ($\ell_2$) | Baseline | $7.60 \pm 0.16$ | $10.05 \pm 0.41$ | $12.47 \pm 0.73$ | $15.42 \pm 1.13$ | $22.19 \pm 3.06$ |
| | SuperLoss | $\mathbf{7.24 \pm 0.47}$ | $\mathbf{8.35 \pm 0.17}$ | $\mathbf{9.10 \pm 0.33}$ | $\mathbf{11.74 \pm 0.14}$ | $\mathbf{13.91 \pm 0.13}$ |
| smooth-$\ell_1$ | Baseline | $6.98 \pm 0.19$ | $7.40 \pm 0.18$ | $8.38 \pm 0.08$ | $11.62 \pm 0.08$ | $17.56 \pm 0.33$ |
| | SuperLoss | $\mathbf{6.74 \pm 0.14}$ | $\mathbf{6.99 \pm 0.09}$ | $\mathbf{7.65 \pm 0.06}$ | $\mathbf{9.86 \pm 0.27}$ | $\mathbf{13.09 \pm 0.05}$ |

**Lemma 2.** *The SuperLoss is a homogeneous function, i.e. $SL^*_{\gamma\lambda}(\gamma\ell) = \gamma SL^*_\lambda(\ell), \ \forall \gamma > 0$.*

*Proof.* Due to the fact that $\sigma^*$ only depends on the *ratio* $(\ell - \tau)/\lambda$ (see Eq. 10) and that $\tau$ is proportional to $\ell$ because it is computed as a running average of $\ell$, we have:

$$\sigma^*_{\gamma\lambda}(\gamma\ell) = \sigma^*_\lambda(\ell).$$

It naturally follows that

$$
\begin{aligned}
SL^*_{\gamma\lambda}(\gamma\ell) &= \sigma^*_{\gamma\lambda}(\gamma\ell)(\gamma\ell - \gamma\tau) + \gamma\lambda(\log\sigma^*_{\gamma\lambda}(\gamma\ell))^2 \\
&= \sigma^*_\lambda(\ell)(\gamma\ell - \gamma\tau) + \gamma\lambda(\log\sigma^*_\lambda(\ell))^2 \\
&= \gamma\left(\sigma^*_\lambda(\ell)(\ell - \tau) + \lambda(\log\sigma^*_\lambda(\ell))^2\right) \\
&= \gamma SL^*_\lambda(\ell). \qquad\qquad\square
\end{aligned}
$$

# 3 Detailed results

## 3.1 Regression

**Hyper-parameters.** During cross-validation of the hyper-parameters, we found that it is important to use a robust error metric to choose the best parameters, otherwise noisy predictions may have too much influence on the results. We thus use a truncated absolute error $\min(t, |y - \hat{y}|)$ where $y$ and $\hat{y}$ are the true value and the prediction, respectively, and $t$ is a threshold set to 1 for MNIST and 10 for UTKFace.

**Detailed results.** For the sake of comparison, we provide detailed experimental results for digit regression and human age regression in Table 1 and Table 2, respectively.

## 3.2 Classification

**Detailed results.** In our experiments on the CIFAR-10 and CIFAR-100 datasets, we have compared to the state of the art under different proportions of corrupted labels. More specifically, we used what is commonly defined as symmetric noise, *i.e.* a certain proportion of the train labels are replaced by other labels drawn from a uniform distribution. We now provide detailed results for the two following cases: (a) the new (noisy) label can remain equal to the original (true) label; and (b) the new label is drawn from a uniform distribution that exclude the true label. We report the results for each case respectively in Table 3 and 4.

Table 3: Detailed results on CIFAR-10 and CIFAR-100 for different proportions of symmetric noise (original labels cannot be maintained).

| | CIFAR-10 | | | CIFAR-100 | | |
|---|---|---|---|---|---|---|
| Method | 20% | 40% | 60% | 20% | 40% | 60% |
| Self-paced [8] (2010) | 89.0 | 85.0 | - | 70.0 | 55.0 | - |
| Focal Loss [12] (2016) | 79 .0 | 65.0 | - | 59.0 | 44.0 | - |
| MentorNet DD [5] (2017) | 91.23 | 88.64 | - | 72.64 | 67.51 | - |
| Forgetting [1] (2017) | 78.0 | 63.0 | - | 61.0 | 44.0 | - |
| Co-Teaching [4] (2018) | 87.26 | 82.80 | - | 64.40 | 57.42 | - |
| $L_q$ [29] (2018)` | - | 87.13 | 82.54 | - | 61.77 | 53.16 |
| Trunc $L_q$ [29] (2018)` | - | 87.62 | 82.70 | - | 62.64 | 54.04 |
| Reweight [21] (2018) | 86.9 | - | - | 61.3 | - | - |
| D2L [14] (2018) | 85.1 | 83.4 | 72.8 | 62.2 | 52.0 | 42.3 |
| Forward $\hat{T}$ [17] (2019) | - | 83.25 | 74.96 | - | 31.05 | 19.12 |
| SELF [2] (2019) | - | 93.70 | 93.15 | - | 71.98 | 66.21 |
| Abstention [25] (2019) | 93.4 | 90.9 | 87.6 | 75.8 | 68.2 | 59.4 |
| CurriculumNet [24] (2019) | 84.65 | 69.45 | - | 67.09 | 51.68 | - |
| O2U-net(10) [6] (2019) | 92.57 | 90.33 | - | 74.12 | 69.21 | - |
| O2U-net(50) [6] (2019) | 91.60 | 89.59 | - | 73.28 | 67.00 | - |
| DivideMix [10] (2020) | 96.2 | 94.9 | 94.3 | 77.2 | 75.2 | 72.0 |
| CurriculumLoss [13] (2020) | 89.49 | 83.24 | 66.2 | 64.88 | 56.34 | 44.49 |
| SuperLoss, $\tau = \log C$ | $93.31 \pm 0.19$ | $90.99 \pm 0.19$ | $85.39 \pm 0.46$ | $75.54 \pm 0.26$ | $69.90 \pm 0.24$ | $61.01 \pm 0.25$ |
| SuperLoss, $\tau$=Avg | $93.16 \pm 0.17$ | $91.05 \pm 0.18$ | $85.52 \pm 0.53$ | $75.02 \pm 0.08$ | $71.06 \pm 0.17$ | $61.96 \pm 0.11$ |
| SuperLoss, $\tau$=ExpAvg | $92.98 \pm 0.11$ | $91.06 \pm 0.23$ | $85.48 \pm 0.13$ | $74.34 \pm 0.26$ | $70.96 \pm 0.24$ | $62.39 \pm 0.17$ |

Table 4: Detailed results on CIFAR-10 and CIFAR-100 for different proportions of symmetric noise (original labels can be maintained).

| | CIFAR-10 | | | | CIFAR-100 | | | |
|---|---|---|---|---|---|---|---|---|
| Method | 20% | 40% | 50% | 80% | 20% | 40% | 50% | 80% |
| Bootstrap [20] (2015) | 86.8 | - | 79.8 | 63.3 | 62.1 | - | 46.6 | 19.9 |
| F-correction [16] (2017) | 86.8 | - | 79.8 | 63.3 | 61.5 | - | 46.6 | 19.9 |
| Mixup [28] (2018) | 95.6 | - | 87.1 | 71.6 | 67.8 | - | 57.3 | 30.8 |
| Co-teaching+ [27] (2019) | 89.5 | - | 85.7 | 67.4 | 65.6 | - | 51.8 | 27.9 |
| P-correction [26] (2019) | 92.4 | - | 89.1 | 77.5 | 69.4 | - | 57.5 | 31.1 |
| Meta-Learning [11] (2019) | 92.9 | - | 89.3 | 77.4 | 68.5 | - | 59.2 | 42.4 |
| Data Parameters [23] (2019) | - | $91.10 \pm 0.70$ | - | - | - | $70.93 \pm 0.15$ | - | - |
| DivideMix [10] (2020) | 96.1 | - | 94.6 | 93.2 | 77.3 | - | 74.6 | 60.2 |
| SuperLoss, $\tau = \log C$ | $93.39 \pm 0.12$ | $91.73 \pm 0.17$ | $90.11 \pm 0.18$ | $77.42 \pm 0.29$ | $74.76 \pm 0.06$ | $69.89 \pm 0.07$ | $66.67 \pm 0.60$ | $37.91 \pm 0.93$ |
| SuperLoss, $\tau$=Avg | $93.16 \pm 0.17$ | $91.55 \pm 0.18$ | $90.21 \pm 0.22$ | $76.69 \pm 0.60$ | $74.73 \pm 0.17$ | $71.05 \pm 0.08$ | $67.84 \pm 0.25$ | $36.40 \pm 0.09$ |
| SuperLoss, $\tau$=ExpAvg | $93.03 \pm 0.12$ | $91.70 \pm 0.33$ | $89.98 \pm 0.18$ | $77.49 \pm 0.29$ | $74.65 \pm 0.31$ | $70.98 \pm 0.26$ | $67.21 \pm 0.33$ | $36.45 \pm 0.80$ |

As mentioned in the paper, the SuperLoss performs on par or better than most of the recent state-of-the-art approaches, including ones specifically designed for classification and requiring dedicated training procedure. We find that SELF [2] and DivideMix [10] are nevertheless outperforming our SuperLoss. Yet we note that they share the aforementioned limitations as both rely on ensembles of networks to strongly resist memorization. In contrast, our approach uses a single network trained with a baseline procedure without any special trick.

**Impact of the regularization.** We show in Figure 2 the impact of the regularization parameter $\lambda$ on CIFAR-10 and CIFAR-100 for different proportions of label corruption. We observe that, overall, the regularization has a moderate impact on the classification performance. At the exception of very high level of noise (80%), the performance plateaus for a relatively large range of regularization values. Importantly, the optimal value of $\lambda$ is approximately the same for all noise levels, indicating that our method can cope well with the potential variance of training sets in real use-cases.

### 3.3 Object Detection

**Hyper-parameters.** For the baseline, we have used the default parameters from detectron2. For the SuperLoss, we have used $\lambda = 1$ for clean data and $\lambda = 0.25$ for any other level of noise, in all experiments, for both Faster R-CNN and RetinaNet.

**Detailed results.** In the main paper, we report the AP50 metric on Pascal VOC for the baseline, the SuperLoss and the state of the art, with varying level of label noise. In Table 5 of this supplementary, we report also the AP75 metric (*i.e.* the mean average precision (mAP) at a higher intersection-over-union (IoU) threshold of 0.75 instead of 0.5, as well as the AP metric, which is the average of mAP

Figure 2: Impact of the regularization parameter for different proportions of noise (we used $\tau$=ExpAvg in all experiments).

Table 5: Comparison of our SuperLoss with the baseline and the state of the art for object detection using the AP, AP50 and AP75 metrics on Pascal VOC. AP50 was already reported in the main paper (without the standard deviation). For the baseline and the Superloss, we report the mean and standard deviation over 3 runs. † denotes numbers reported from CA-BBC.

| | label noise | AP | | | | AP50 | | | | AP75 | | | |
|---|---|---|---|---|---|---|---|---|---|---|---|---|---|
| | | 0% | 20% | 40% | 60% | 0% | 20% | 40% | 60% | 0% | 20% | 40% | 60% |
| RetinaNet | baseline | $54.9_{\pm0.5}$ | $51.0_{\pm0.2}$ | $48.7_{\pm0.3}$ | $34.9_{\pm0.6}$ | $80.6_{\pm0.2}$ | $77.5_{\pm0.7}$ | $74.6_{\pm0.6}$ | $52.0_{\pm3.3}$ | $59.7_{\pm0.5}$ | $55.6_{\pm0.7}$ | $51.8_{\pm0.5}$ | $36.8_{\pm2.4}$ |
| | SuperLoss $\tau$=ExpAvg | $54.8_{\pm0.6}$ | $52.1_{\pm0.4}$ | $\mathbf{49.8}_{\pm0.1}$ | $\mathbf{39.3}_{\pm0.3}$ | $80.5_{\pm0.4}$ | $\mathbf{78.1}_{\pm0.2}$ | $\mathbf{75.3}_{\pm0.0}$ | $59.6_{\pm0.2}$ | $59.8_{\pm0.7}$ | $56.6_{\pm0.5}$ | $\mathbf{54.3}_{\pm0.3}$ | $42.7_{\pm0.4}$ |
| | SuperLoss $\tau$=Avg | $\mathbf{55.3}_{\pm0.1}$ | $\mathbf{52.3}_{\pm0.4}$ | $\mathbf{49.8}_{\pm0.3}$ | $\mathbf{39.3}_{\pm1.0}$ | $\mathbf{80.7}_{\pm0.2}$ | $78.0_{\pm0.1}$ | $75.2_{\pm0.8}$ | $\mathbf{59.7}_{\pm1.6}$ | $\mathbf{60.6}_{\pm0.2}$ | $\mathbf{56.9}_{\pm0.3}$ | $54.1_{\pm0.4}$ | $\mathbf{42.8}_{\pm1.3}$ |
| Faster R-CNN | baseline | $\mathbf{53.2}_{\pm0.2}$ | $47.3_{\pm0.3}$ | $44.5_{\pm0.3}$ | $41.5_{\pm0.5}$ | $\mathbf{81.4}_{\pm0.0}$ | $76.9_{\pm0.1}$ | $73.6_{\pm0.2}$ | $69.5_{\pm0.4}$ | $\mathbf{58.2}_{\pm0.2}$ | $50.4_{\pm0.8}$ | $46.8_{\pm0.1}$ | $43.2_{\pm0.7}$ |
| | SuperLoss $\tau$=logC | $52.7_{\pm0.1}$ | $\mathbf{50.8}_{\pm0.1}$ | $\mathbf{49.3}_{\pm0.1}$ | $\mathbf{46.6}_{\pm0.2}$ | $\mathbf{81.4}_{\pm0.1}$ | $\mathbf{79.5}_{\pm0.3}$ | $\mathbf{78.1}_{\pm0.1}$ | $\mathbf{74.9}_{\pm0.1}$ | $57.8_{\pm0.2}$ | $\mathbf{55.4}_{\pm0.2}$ | $\mathbf{53.6}_{\pm0.2}$ | $50.0_{\pm0.3}$ |
| | SuperLoss $\tau$=ExpAvg | $52.5_{\pm0.2}$ | $49.0_{\pm0.2}$ | $48.5_{\pm0.2}$ | $46.4_{\pm0.3}$ | $81.0_{\pm0.2}$ | $78.4_{\pm0.1}$ | $77.0_{\pm0.3}$ | $73.8_{\pm0.4}$ | $57.4_{\pm0.3}$ | $52.9_{\pm0.3}$ | $52.3_{\pm0.2}$ | $\mathbf{50.2}_{\pm0.5}$ |
| | Co-teaching† | | | | | 78.3 | 76.5 | 74.1 | 69.9 | | | | |
| | SD-LocNet† | | | | | 78.0 | 75.3 | 73.0 | 66.2 | | | | |
| | Note RCNNN† | | | | | 78.6 | 75.3 | 74.9 | 69.9 | | | | |
| | CA-BBC† | | | | | 80.1 | 79.1 | 77.7 | 74.1 | | | | |

at varying IoU thresholds. We report both the mean and the standard deviation over 3 runs. With both Faster R-CNN and RetinaNet object detection frameworks, we observe that the SuperLoss allows to significantly increase the performance of the baseline in the presence of noise for all metrics. Interestingly, the SuperLoss also significantly reduces the variance of the model, which is pretty high in the presence of noise, in particular with RetinaNet.

### 3.4 Image Retrieval

**Hyper-parameters.** As mentioned in the main paper, we use a ResNet-50 with Generalized-Mean (GeM) pooling [19] and a contrastive loss [3][4]. When training on Landmarks-clean, we use the default hyper-parameters from [18] for the optimizer and hard-negative mining. Specifically, we train for 100 epochs using the Adam optimizer with a learning rate and weight decay both set to 1e-6. The learning rate is exponentially decayed by $\exp(-1)$ overall. At each epoch, 2000 tuples are fed to the network in batches of 5 tuples. Each tuple is composed of 1 query, 1 positive and 5 negatives (*i.e.* 1 positive pair and 5 negative pairs, mined from a pool of 20K hard-negative samples).

**Detailed results.** When trained on the noisy Landmarks-full dataset using the same settings, the baseline has trouble to converge and performs poorly, see Figure 3. We believe that this is due to the fact that hard-negative mining systematically finds wrongly labeled negative pairs that prevent the model to properly learn. For this reason, we tune again the learning rate and the hard-negative mining parameters of the baseline on the validation set of Landmarks-full. We find that reducing the size of the negative pool indeed improves the situation as it makes it less likely that noisy negative images are found. The new learning rate, number of tuples and size of the negative pool are respectively 1e-5, 200 and 500. Since the network sees less tuples per epoch (and hence less pairs), we choose to train twice longer (200 epochs instead of 100) with the same overall exponential decay of the learning rate. For the SuperLoss, we use global averaging to compute $\tau$ and validate $\lambda = 0.05$ on the validation set.

Figure 3: Model convergence during training on the noisy Landmarks-full dataset.

The convergence of the re-tuned baseline and the SuperLoss are shown in Figure 3. Even though the baseline drastically improves and now converges properly, its is still outperformed by a large martgin by the SuperLoss at all stages of the training.

**Evaluation metrics.** We have evaluated our approach using the Revisited Oxford and Paris benchmark [18]. It is composed of two datasets, Oxford and Paris, that consist of respectively 5,063 and 6,392 high-resolution images. Each dataset contains 70 queries from 11 landmarks. For each query, positive images are labelled as easy or hard positives. For each dataset, we evaluate in term of mean Average-Precision (mAP) using the medium (M) and hard (H) protocols which consist in respectively considering all positive images or only hard ones (*i.e.* ignoring easy ones). We refer the reader to [18] for a more detailed explanation on the benchmark and the evaluation metrics.

## Footnotes

[1]We give here the correct formula as there is a typo in the formula of the original paper.

[2]we only consider the case with instance parameters for simplicity

[3]For instance, it is available in python as `scipy.special.lambertw`

[4] https://github.com/filipradenovic/cnnimageretrieval-pytorch