[Reviews · NeurIPS 2020]

Review 1

Summary and Contributions: Comment after author response: I thank authors for their response. My concern about the definition of hard and easy examples centers on data points with noisy labels, there is no clear way to distinguish between truly hard examples versus easy examples with noisy labels, and vise versa. This analysis might explain why the model does not considerably improve performance against baselines (at least [47]). In addition, I suggest conducting thorough comparison (including reports on training turnaround time) against CIFAR datasets across all noise levels and against the WebVision dataset. ========================= The paper introduces a new (and to some extent task-agnostic) loss function for curriculum learning-based training of neural networks. The idea is inspired by recent works on confidence-aware loss functions which try to jointly learn model parameters and confidence scores of samples during training, where the confidence score (weight) of each sample determines its contribution to training -- easier (harder) instances are those with higher (lower) confidence scores and contribute more (less) to training. The model computes pseudo-optimal confidence scores for instances using their current loss. The proposed loss function includes a regularization term (Lambda) to handle different levels of noise in training data, and a threshold (tao) to dynamically distinguish hard and easy examples based on their respective loss. The idea is simple and incremental, but it departs from existing works by simplifying the process of employing confidence scores in curriculum learning (confidence scores are directly derived from current loss instead of being learned through backpropagation). In addition, in contrast to most CL-based training models, the proposed loss function does not require a dedicated training process and can be/has been applied (with modification) to many loss functions and tasks without any change in the training algorithm.

Strengths: - The model can handle different levels of noise in training data, and can dynamically distinguishes hard and easy examples based on their respective loss. - The model simplifies existing work that employ confidence scores in curriculum learning and does not require a dedicated training process. - The model be/has been applied to many loss functions and tasks without any change in the training algorithm.

Weaknesses: - The definition of hard and easy examples is limited to their respective confidence scores or losses. - The paper partially illustrates a potential challenge in the current state-of-the-art technique but does not elaborate on it. - The model does not considerably improve performance metrics and is often on par with other approaches. - The paper lacks experiments illustrating turnaround training times of competing models.

Correctness: yes.

Clarity: yes.

Relation to Prior Work: yes. There are a few papers (that use spaced repetition as curriculum) that could be cited.

Reproducibility: Yes

Additional Feedback: Further comments: - The definition of hard and easy examples is limited to their respective confidence scores or losses. Although previous work has similar definitions, confidence or loss are not always good indicators of true easiness or hardness of samples, e.g. they could be erroneous at early iterations. The paper lacks an experiment that illustrates the validity of the above definition. - In Figure 2, there are wrongly predicted samples with high confidence scores (bottom-right). These are probably hard or noisy examples that were mistreated as easy examples by the model? - Again, in Figure 2, lower confidence scores do not necessarily mean "constant" loss. These are probably a mixture of easy, hard, and noisy examples with low confidence across the loss spectrum that were mistreated as hard examples by the model. - Figure 3 (left) illustrates a potential challenge in the current state-of-the-art technique where parameterizing confidence scores during training can induce delay in convergence. However, the paper lacks further analysis to illustrate the effect of such delays. Important questions to ask include: what's the extent of delay across all training samples, and what's the effect of decreasing/increasing such delays by changing related hyperparameters. I think illustrating the importance of this challenge can be an important contribution of the paper. - Experiments: in terms of performance, the model does not considerably improve performance metrics and is often on par with other approaches. Perhaps this is because the model does not properly distinguish easy from hard examples. - Experiments: the paper lacks experiments illustrating turnaround training times of competing models. - Experiments: how similar/different are the set of easy and hard examples acorss different runs of the model? - For performance improvement, I wonder if spaced repetition (a curriculum that starts with entire training data and gradually delays training of easier examples) could be used for more effective training? see Amiri, et al., Repeat before forgetting: Spaced repetition for efficient and effective training of neural networks, EMNLP 2017, https://www.aclweb.org/anthology/D17-1255/


Review 2

Summary and Contributions: This method presents a general method for re-weighting samples through training (ie. curriculum learning) to improve performance and robustness. The method revolves around a confidence aware loss function, the SuperLoss, which introduces a confidence parameter sigma_i per sample (x_i, y_i) and can be applied on top of any loss function. Due to the choice of the SuperLoss function, the optimal sigma_i can be derived in closed form for any value of the loss function; thus in practice the SuperLoss function effectively re-weights samples during training, up-weighting samples with low loss (high confidence) and down-weighting samples with high loss (low confidence). The reweighting is applied across a range of tasks: classification, regression, object detection, and retrieval; and on a range of datasets: MNIST, UTKFace, CIFAR-10, CIFAR-100, WebVision, Pascal, and Landmarks. The experiments are thorough and the results are positive in every domain, although not outstanding in any particular one. The method is particularly effective at training with noisy labels, as it drops in performance significantly less than many prior works.

Strengths: The main strength is the robustness of the results and the simplicity of the method. This is a component that anyone doing deep learning can add to their training at little to no cost, since the confidence parameter can be computed in closed form unlike some prior work.

Weaknesses: The paper is missing a probabilistic explanation of the loss function (equation 1). Given that it is in fact very similar to prior re-weighting functions that have been used (figure 1), it it somewhat surprising that it does perform much better than the previous ones. Appendix section 1 describes several similar losses - understanding what graphical model underlies each one may give a better path to designing better curriculum weights, rather than picking yet another one.

Correctness: No issues with the methodology.

Clarity: The writing is relatively well-written, organized, clear, and easy to read. Appendix eqn 10: Just to confirm, you do not have derivatives for sigma^* right? Ie. sigma^* is treated as a constant and d sigma / d l is not considered for back-propagation?

Relation to Prior Work: Yes, this is clearly discussed and in fact is another strength of the paper that it describes a range of prior work that were proposed for different problems in the same framework.

Reproducibility: Yes

Additional Feedback: == After rebuttal == Keeping my score the same. The algorithmic novelty is small compared to the nearest neighbor works. However there are definitely several positives. The empirical results seemed relatively strong (although the other reviewers raised good concerns about some of the details): they are on a wide variety of tasks, and at least in some cases seems to provide a significant performance boost for little cost/effort (particularly with noisy labels). The paper is thorough in comparing many prior methods, and also presents the prior methods which are somewhat domain-specific in a nice, general way. And the method is really quite general and easy to implement.


Review 3

Summary and Contributions: This paper proposes a sample-weighting method for robust curriculum learning when training a model with noisy labels. They design a sample-weighted loss called SuperLoss that leads to a confidence score (i.e. the sample weight) that downweighs the hard samples with large loss values and emphasizes the easy samples with small loss values. This re-weighting idea has been shown effective in previous works since the noisy labeled data usually suffer from a large loss while the clean data are usually associated with a small loss. The advantage of SuperLoss compared to existing confidence-aware scores, as claimed in the paper, is that the scores are optimal w.r.t. the defined SuperLoss objective in Equ.1 and have a closed-form solution, while the learnable scores in existing works need a relatively delayed process of convergence to reach its optimal values during training. This is shown in Fig. 3. They also claim that the SuperLoss can be applied to any loss in arbitrary tasks. Experiments on several tasks and benchmark datasets show that SuperLoss is not very helpful in the clean data setting, but can bring improvement in noisy labeled data setting. However, in the noisy label setting, the gap between SuperLoss's performance and the best noisy-label learning baseline's performance is small in most experiments.

Strengths: 1) The SuperLoss reflects a broadly-supported strategy in noisy-label learning, i.e., down-weighting the hard samples with large loss can mitigate the influence of noisy data. 2) The experiments cover different tasks and benchmark datasets. The SuperLoss shows reasonable performance.

Weaknesses: 1) Different from existing works of learnable sample weights, the objective in Equ.1 used to optimize the weights in this paper is hand-crafted without sufficient justification: what is the relationship to the training loss or the validation (held-out set or meta-set) loss? How does this objective help to improve the model performance theoretically? What is the motivation of having the second term in Equ.1? According to the current draft, I lean to believe that the objective is reverse-engineered for the closed-form solution, which reflects the widely-accepted intuition of down-weighting hard samples for noisy label learning. However, compared to existing works that explicitly define the objective of optimizing weights as minimizing the tilted training loss or held-out set's loss, the objective in this paper is less motivated. 2) It is misleading to claim that the paper is the first work using task-agnostic weights that do not require iterative learning, since many curriculum learning methods using loss-based sample-weights, e.g., SPL and its variants, do not require the learning of the weights and thus are all task-agnostic without requiring iterative learning. The potential problem of these methods is that their scoring criteria might be too intuitive since they are not explicitly derived from any meta-objective for the trained model. Although this paper's weights are derived from a hand-crafted objective, it is not clear how the objective relates to the final performance of the trained model. Hence, it is not fair to compare it to those who do, e.g., in Fig.3, since other deterministic weights also have such an advantage over the iteratively learned weights. 3) Although the proposed SuperLoss is novel in its form, it does not brings notably new criteria in determining the sample weights, as mentioned before. It results in the same trend of down-weighting hard samples but nothing more. So the potential contribution is very limited. 4) I appreciate the diversity of the tasks in the experiments, but the comparison is not always thorough on different tasks. For example, the baselines for clean data experiments are too few, i.e., 1-2 baselines that ignore many existing curriculum learning methods applicable to clean-data setting. SuperLoss does not show an advantage on clean data even comparing to conventional training without a curriculum. 5) In noisy label setting, in specific, Fig. 6 and the right table in Fig. 7, SuperLoss does not show an overwhelming advantage over the baselines. In Fig. 6, its curve overlaps with several baselines. In the right table of Fig. 7, the maximal gap between SuperLoss and the best baseline is < 1%. Therefore, it is hard to justify that SuperLoss is powerful in the noisy label scenario. 6) Using the loss or the confidence to split noisy/clean data has potential risks when training DNN since DNN can be easily overconfident on wrongly-labeled data, especially the data labels contain a lot of noises. The proposed SuperLoss essentially solely depends on the thresholded loss, which is problematic for the same reason. I suggest the authors conduct a detailed empirical or theoretical analysis of this problem.

Correctness: The techniques are correct under its assumption, but the motivation and comparison statement to existing works are misleading. The objective in Equ.1 lacks sufficient justification and is likely to be reverse-engineered for the closed-form solution.

Clarity: The writing is overall clear, but modifications to a variety of statements in this paper are necessary.

Relation to Prior Work: The paper's discussion covers most related works, but the discussion of their difference to the proposed method is biased and misleading to some extent (please see weakness 1 & 2).

Reproducibility: Yes

Additional Feedback: The paper's discussion covers most related works, but the discussion of their difference to the proposed method is biased and misleading to some extent (please see weakness 1 & 2). ==============post rebuttal============== Thanks for the rebuttal! However, there are several fundamental issues about the empirical comparison and the novelty compared to previous methods after considering the rebuttal: - In Fig.6, SuperLoss does not outperform several baselines. On CIFAR100, the accuracy of SuperLoss quickly degrades below several baselines as the noisy level increases, while the high noise region is the main challenge of noisy-label learning. - The experiments only compare SuperLoss to [47] at one single noise level, i.e, 30-40%, which is much lower than mainstream noisy-label learning papers did. - The curves in Fig.6 are not accurate enough to reflect the advantage of SuperLoss, since each curve is fitted based only on <3 points. - SuperLoss shares the same idea as previous methods but nothing notably new, i.e., placing more training on samples with smaller loss. The major difference is on their choices of a monotone decreasing function of loss (Eq.(10) for SuperLoss). But this paper does not provide sufficient justification on the objective Eq.(1) defining the monotone function, nor explaining why their choice is better than the other monotone decreasing functions. Hence, I think this paper is not ready for publication yet.


Review 4

Summary and Contributions: This paper contributes a new loss formulation, the "SuperLoss", that dynamically modulates the contributions of a primary task loss during training. In essence, it down-weights high loss values in the style of curriculum learning. The proposed SuperLoss is compared to existing approaches on the tasks of image classification, regression, object detection and image retrieval, and shown to perform favourably.

Strengths: 1. The loss formulation is generic and task-agnostic and therefore has the potential for broad applicability. The experimental evidence, collected over a range of tasks indicate that the loss contributes particular value in the (ubiquitous) setting in which a portion of the annotations are noisy. 2. In comparison to many curriculum learning strategies, which often require a number of carefully selected hyperparameters (relating to the timing of the schedule and the weights that should be applied at various times), the SuperLoss performs well with a single weighting function (to which, as the results show in Section 3, the loss is not highly sensitive). 3. The authors provide a nice intuition for how losses evolve over time (Fig 3, left), justifying the choice of automatic confidence selection (vs potentially stale learned confidences).

Weaknesses: The experimental evaluation shows that the SuperLoss can indeed be applied to multiple tasks (which is a strength). But I would not consider any of these evaluations to be particularly large-scale (so there is room for improvement in the statistical strength of the results). The exceptions to this statement are (i) Webvision, in which SuperLoss is compared only to standard Cross Entropy (if I understand the description in Sec. 3.3 correctly) and (ii) the Image Retrieval benchmark considered in 3.5, but here the proposed SuperLoss is compared only to a vanilla contrastive loss. By contrast, the prior work of [47] (citation number from submission) evaluated their approach on ImageNet, which provides greater statistical strength. This factor was my primary reason for not assigning a higher score to the paper.

Correctness: The claims of effectiveness are validated empirically. The methodology seems reasonable and correct.

Clarity: The paper is well written and easy to follow.

Relation to Prior Work: The contribution is clearly explained in the context of prior work. Figure 2, in particular, gives a nice intuition for how prior "confidence-aware" losses differ from each other.

Reproducibility: Yes

Additional Feedback: Update (following the rebuttal and discussions with other reviewers). After reading the rebuttal and discussing this paper with the other reviewers, I have slightly lowered my score. The reasons were: (i) concerns about the novelty of the approach, highlighted by other reviewers, (ii) the empirical evaluation, which does not seem watertight. Specifically, in addition to the concerns raised by R3, the submission does not include comparisons with [47] on WebVision (which are available from [47], and show the performance gains to be similar) or comparisons on object detection (the submission gives the rationale that this would be infeasible with [47], but [47] contains experiments on detection with KITTI, so a comparison would have been possible).

[Author Response · NeurIPS 2020]

We thank the reviewers for their feedback. Our 'formulation is generic and task-agnostic and therefore has the potential
for broad applicability' (**R4**). 'The main strength is the robustness of the results and the simplicity of the method' (**R2**).
'The model simplifies existing work' (**R1**) and 'has been applied to many loss functions and tasks without any change
in the training' (**R1**). 'The experiments cover different tasks and benchmark datasets' (**R3**).

**'It is misleading to claim that the paper is the first work using task-agnostic weights that do not require iterative**
**learning' (R3.2).** *We do not make such a claim.* What we claim is: in contrast to SPL variants and any other related
work, our approach applies directly to any baseline without any change whatsoever in the training procedure other than
plugging the SuperLoss on top (lines 51-57). We believe a simple and easy-to-use idea has potential for great impact.

**Motivation for Equation 1 (R3.1). 'Missing a probabilistic explanation' (R2).** While it is true that our SuperLoss is
not derived from probabilistic considerations, we point out that our approach for designing the SuperLoss *is* principled
and insightful for future work. Namely, we establish a connection between curriculum learning and a family of loss
functions that we denote as confidence-aware (CA). We review (in Section 2.1 and Section 1 from the supplementary)
existing CA losses and study for the first time their properties, in particular their gradient monotonicity (*i.e.* the
gradient of the loss *w.r.t.* the network parameters monotonously increases with the confidence) which is at the root of
their connection to dynamic curriculum learning. At the same time, we also emphasize their different shortcomings:
unfortunately none of the existing CA losses is task-agnostic (Suppl. Section 1). We therefore propose in Section 2.2 the
first *generic* CA formulation, *i.e.* that is jointly able to (1) handle losses of any scale (*i.e.* it is homogeneous, see Suppl.
Section 2.3); (2) handle both positive- and negative-valued losses (which justifies the squared regularizer log term[1]);
and (3) generalize the input loss (i.e. $SL^*(\ell) \to \ell$ when $\lambda \to \infty$, Suppl. Section 2.2). Our SuperLoss is also easily
interpretable since $\sigma^*$ directly corresponds to the sample weights (Suppl. Section 1). On top of that, our formulation is
among the simplest possible ones that achieves all these properties simultaneously. We agree that the exposition of the
design process of the SuperLoss can be improved and will update Section 2.2 and the rest of the paper accordingly.

**'Does not brings notably new criteria in determining the sample weights' (R3.3).** Compared to other methods that
learn sample weights via backpropagation in a unified framework like [47], our formulation goes one step further and
directly uses the converged value of the weights according to the current state. We thereby avoid additional parameters,
hyper-parameters and inconsistencies due to delay (Section 2.3 and Figure 3 left). We obtain better results in fair settings
compared to learning the weights via backpropagation (comparison with DataParameters in Table 4 from Suppl.).

**'SuperLoss does not show an advantage on clean data' (R3.4).** On clean data, the difference is indeed marginal, as
for other curriculum learning methods on such standard datasets with recent deep residual networks.

**'In noisy label setting [...] its curve overlaps with several baselines' (R3.5). 'The model does not considerably**
**improve performance metrics and is often on par with other approaches' (R1).** Other methods are often limited to
a single loss like cross-entropy while our SuperLoss applies to any loss and is easy-to-use (about 10 lines of new code).

**'DNN can be easily overconfident on wrongly-labeled data (R3.6)'.** That is true. However, we observe in practice
that noisy samples are well separated from clean samples even under heavy noise, see Figure 6 where noisy samples are
clearly downweighted at the end of the training. We will study the extent of this resilience more precisely in the paper.

**Large-scale experiments (R4).** For classification, WebVision is larger than ImageNet (2.4M images) and additionally
contains some noise due to the automatic collection process. We also provide retrieval experiments on Landmarks,
where for the first time we show that training a model on the full dataset (160K images) perform better than training on a
subset of automatically verified images. This is a strong large-scale result which shows that large datasets automatically
collected can be used instead of manually annotated ones.

**'The definition of hard and easy examples is limited to their respective confidence scores' (R1).** We indeed follow
related work in this definition. We are not aware of any large-scale dataset with annotations for the difficulty of
samples, and we point that human annotations might actually not correspond to the actual difficulty from a deep model
perspective. We separate hard from easy samples at epoch 20, which is already quite significant. About Figure 2, our
SuperLoss does not use the full spectrum of the plot as the optimal confidence value $\sigma^*$ depends on the input loss, thus
avoiding samples with low loss (correct prediction) and low confidence by design.

**'The paper lacks experiments illustrating turnaround training times of competing models.' (R1)** In practice, the
observed overhead in training time for SuperLoss on CIFAR100 is 0.4% longer than for the corresponding baseline.

**'$\sigma^*$ treated as constant' (R2).** We confirm that and will make it clear.

**'The paper partially illustrates a potential challenge in the current state-of-the-art technique but does not elab-**
**orate on it' (R1).** Thanks for the suggestion, we agree it would be interesting to look deeper into this direction.

## Footnotes

[1]For negative-valued losses $\ell < 0$, the probabilistic formulation from [21] ($\ell/\sigma + \log \sigma$) blows up to $-\infty$ when $\sigma \to 0$ whereas $\ell/\sigma + (\log \sigma)^2$ (our SuperLoss) behaves suitably, see Section 1 and the closed-form solution in Section 2.1 from the supplementary.


[Meta-Review · NeurIPS 2020]

All reviewers mention that despite the idea being incremental, it is simple and yet effective. The empirical evaluation is made on a number of problems and in that sense is quite thorough. However, at the same time some reviewers have raised concerns about the experimental setup (R3, R4). R3 has two major concerns: limited novelty and improper empirical validation. R3 says, "... results in the same trend of down-weighting hard samples but nothing more. So the potential contribution is very limited". In terms of results, ".. On CIFAR100, the accuracy of SuperLoss quickly degrades below several baselines as the noisy level increases ..". Given that proposed method applies to many different problems (as shown by experiments), I am not concerned about novelty. In their rebuttal, authors acknowledge that their method works is at par-in noisy scenarios with baselines, however as opposed to previous methods that are specific to one particular loss function, their method can be applied to many different loss-functions. This is reasonable. R4's main concern was that statistical-significance of results is not high as no evaluation is made on Imagenet. Authors do provide results on Object Detection on PASCAL VOC, so I am not concerned about this. In my opinion, authors have addressed R4's main concern and mostly addressed R3. At the same time, I would encourage authors to report a thorough comparison with [47] in the main paper for the camera ready version. With this, I believe this is a fine piece of work to be presented to the NeurIPS community.